# Simulation of Ice-Propeller Collision with Cohesive Element Method

**Li Zhou [1], Feng Wang [2], Feng Diao [3], Shifeng Ding [1,*], Hao Yu [4] and Yang Zhou [4]**

[1] School of Naval Architecture and Ocean Engineering, Jiangsu University of Science and Technology, Zhenjiang 212003, China

[2] School of Naval Architecture, Ocean and Civil Engineering, Shanghai Jiao Tong University, Shanghai 200240, China

[3] Shanghai branch, China Ship Scientific Research Center, Shanghai 200011, China

[4] Chin Shipbuilding Industry Corporation Limited, Beijing 100044, China

[*] Correspondence: 15001945469@163.com

**Abstract:** The existence of ice in ice-covered waters may cause damage to the propeller of polar ships, especially when massive ice floes are submerged around the hull. This paper aims to simulate an interaction process of a direct ice collision with a propeller based on the cohesive element method. A constitutive law is applied to model the ice material. The model of ice material is validated against model test results. The resulting impact loads acting on the contact surfaces and the corresponding ice block velocity are calculated in the time domain. The ice crushing, shearing and fracture failures are reproduced in the simulation. The convergence study with three meshing sizes of ice block is performed. To carry out a parametric study, five parameters are selected for analysis. These parameters are composed of rotational speed, direction of the propeller, initial speed of the ice block, contact position, and area between the ice and the propeller. The results show that the ice loads are affected by the five factors significantly. Ice loads tend to increase by decreasing the rotational speed, increasing the initial ice speed and the contact area, and changing the rotational direction from clockwise to counterclockwise. The effect of the contact position on the impact loads is relatively complex, depending on rotational speeds of the propeller.

**Keywords:** propeller-ice collision; ice loads; cohesive element; ice failure;

## 1. Introduction

Arctic shipping has become increasingly attractive due to the huge economic value compared to traditional sea routes. Polar ships or ice-strengthened ships are expected to navigate in the ice-covered seas safely and efficiently. However, sea ice as a solid material may endanger the hull structure and the propeller. For hull design, some regulations or codes are used to ensure that the strength of the hull is high enough to resist ice impact. As for the propeller, this often interacts with underwater ice floes, which is not easily observed in different realistic operation modes. The resulting ice load of dynamic ice-propeller interaction is not clear and should be investigated further.

In general, there are two kinds of ice-propeller interactions. The first one is indirect interaction. An ice block keeps away from the propeller and does not contact the propeller directly. It only affects the flow field around the propeller and thus makes propulsion efficiency decline to some degree. The second one is direct interaction. The propeller may mill or collide with an ice block depending on many factors such as the contact position, the size of ice block, the movement of the ice block, the rotational speeds of the propeller, and so on.

Direct propeller-ice interaction has been studied in many ways. For the ice-propeller milling process, the ice block remains still and the propeller moves towards the ice with a certain rotational speed. Some scale model tests of propeller milling with ice were performed to measure ice loads [1–5]. The numerical simulation for ice-propeller milling were also tried. Wang et al. [6,7] applied a numerical method to predict ice milling load. Ye et al. [8] and Wang et al. [9] simulated an ice-strengthened propeller-ice milling process and analyzed the main factors that may affect the ice load significantly with peridynamics theory.

Another type of direct interaction is called an ice-propeller collision, under which ice block moves freely in 6 degrees of freedom (dofs) during the interaction. Brouwer et al. [10] tried a model test to measure ice impact on a propeller in 6 dofs. However, it is hard to implement the test since main factors influencing the load were not easily controlled. Simple theoretical methods were used to calculate the impact loads during the collision [11,12]. The limitation is that dynamic rotation of the propeller and the full contact of blades with an ice block were not considered in the calculation. As the development of computation, numerical simulations have gradually been used for propeller–ice collision analysis with some commercial software. Khan et al. [13] applied a numerical method to study the impact load for an ice-propeller collision. The method was also validated with the measured results from experiments. Hu [14] developed a propeller–ice collision model in the commercial software LS-DYNA (Livermore Software Technology Corporation, Livermore, CA, USA). The ice block model was built by the smoothed particle hydrodynamic (SPH) method while the blade model was made by the finite element Lagrange method. The results show that it is difficult to reproduce the rational fracture and damage of an ice block.

The mechanism of ice-propeller interaction is not well understood so far. To give insight into the collision process, a rational propeller–ice collision model is built by using the cohesive element method to simulate the interaction. The structure of the propeller is assumed to be rigid and the ice block could move freely during the collision. The phenomenon of different ice failure modes is reproduced with simulated ice loads in the time domain. The convergence and parametric study are performed to analyze the main parameters that would affect the impact of ice loads.

## 2. Numerical Model

The failure patterns of ice include crushing and fracture when the ice blocks hit on the propeller blade. The traditional finite element method (FEM) method is based on the continuous medium hypothesis, which finds it difficult to solve these dis-continuous problems. Thus, cohesive element method (CEM) is introduced to solve the occurrence of ice fracture during the interaction between ice and propeller.

The CEM is developed from the cohesive zone model (CZM), which is one of the FEM simulations. Hillerborg et al. [15] firstly proposed CZM to simulate the crack propagation on an unreinforced concrete beam. Subsequently, many researchers used CEM for simulating the fracture of sea ice [16–18]. CEM showed good performance in solving those questions. There are 2 kinds of elements to model ice in CEM, bulk elements and cohesive elements (see Figure 1). Bulk elements are used to simulate the ice block while the cohesive elements are used to construct the internal faces between two neighboring bulk elements [19–22].

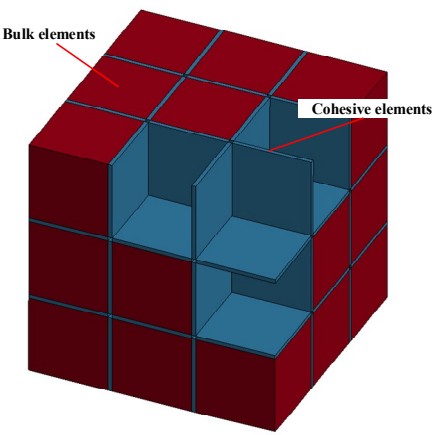

**Figure 1.** Element types of cohesive element method (CEM).

In CEM simulation, linear softening model is a traction-separation law which is the most commonly applied. The corresponding function is written as follows [23],

$$T = \begin{cases} \dfrac{T_0}{\delta_n^1}\delta_n & 0 \le \delta_n < \delta_n^1 \\[2mm] \dfrac{T_0}{(\delta_n^f - \delta_n^1)}(\delta_n^f - \delta_n) & \delta_n^1 \le \delta_n < \delta_n^f \\[2mm] 0 & \delta_n \ge \delta_n^f \end{cases} \tag{1}$$

$$S = \begin{cases} \dfrac{S_0}{\delta_\tau^1}\delta_\tau & 0 \le \delta_\tau < \delta_\tau^1 \\[2mm] \dfrac{S_0}{(\delta_\tau^f - \delta_\tau^1)}(\delta_\tau^f - \delta_\tau) & \delta_\tau^1 \le \delta_\tau < \delta_\tau^f \\[2mm] 0 & \delta_\tau \ge \delta_\tau^f \end{cases} \tag{2}$$

where $T$ and $S$ denote cohesive forces in normal and tangential direction; $\delta_n$ and $\delta_\tau$ denote crack separations in normal and tangential direction; $T_0$ and $S_0$ denote facture stress in normal and tangential directions; $\delta_n^1$ and $\delta_\tau^1$ denote corresponding crack separations; $\delta_n^f$ and $\delta_\tau^f$ denote maximum crack separations.

The relationship between normal separation and cohesive force is presented in Figure 2. For mode I fracture, cohesive force $T$ increases linearly with the growth of crack separation $\delta_n$ under impact at the first stage. When it reaches up to maximum stress $T_0$, the crack starts to occur from this moment. As $\delta_n$ continues to increase, the ice stiffness shows linear softening with expansion of the crack. When $T$ drops to zero, maximum crack separations $\delta_n^f$ is achieved, where the cohesive element fails with adjacent bulk elements separated. The fracture energy GIC is released during this process, which is calculated as the area of the traction-separation curve.

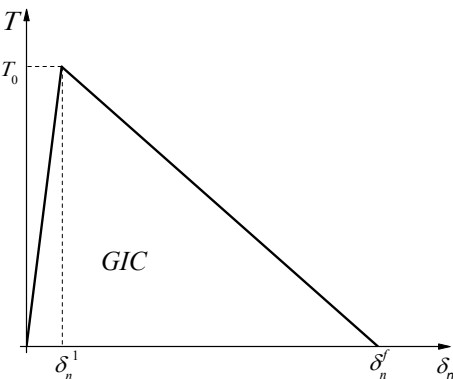

**Figure 2.** Relationship between cohesive force and crack separation for model I fracture.

Local crushing should be taken into consideration in the ice-propeller interaction process as well in addition to fracture and crack. Usually, mesh size with FEM simulation is so large that it cannot capture the local crushing due to the difficulty of explicitly modelling microscopic failure. A constitutive law based on elastoplastic linear softening was used and has been validated effectively [18,20]. The corresponding hardening curve is given in Figure 3, where $\sigma_Y$ is compressive strength as initial crushing point. Crushed strain $\varepsilon_c$ is derived after going through a linear softening phase. The ice material could be taken as completely crushed afterwards. When it advances to failure strain $\varepsilon_f$, the ice element fails and is deleted from the original ice block. This constitutive law is used in the present simulation.

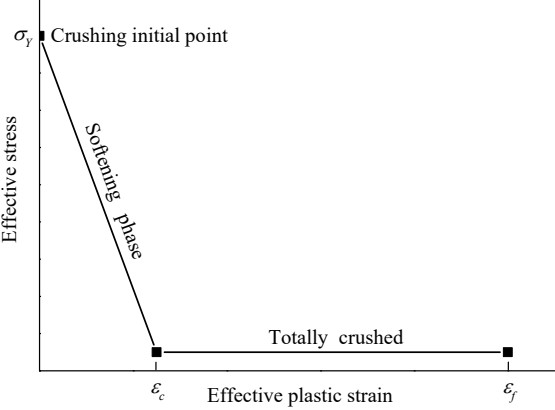

**Figure 3.** Elastoplastic constitutive law given.

## 3. Numerical Model of Ice-Propeller Collision

Figure 4 shows a full-scale ice-propeller collision model made with software LS-DYNA. The propeller will rotate at a constant speed $\omega_P$ = 2rps anticlockwise, while the ice block will move towards the back of the blade along the negative direction of the x-axial. The blades were designed with reference to Stone Marine Meridian series [23–25]. There were ice-strengthened by increasing the thickness for navigation in ice. There are 4 blades in total. The diameter for each blade is 4.12 m. The mean-pitch/diameter ratio and expanded blade area ratio are 0.76 and 0.669, respectively. The structural model of the propeller and its meshing are presented in Figure 5. It is assumed that no deformation occurs to the propeller under ice block impact. The material of the propeller is taken as rigid steel with properties given in Table 1.

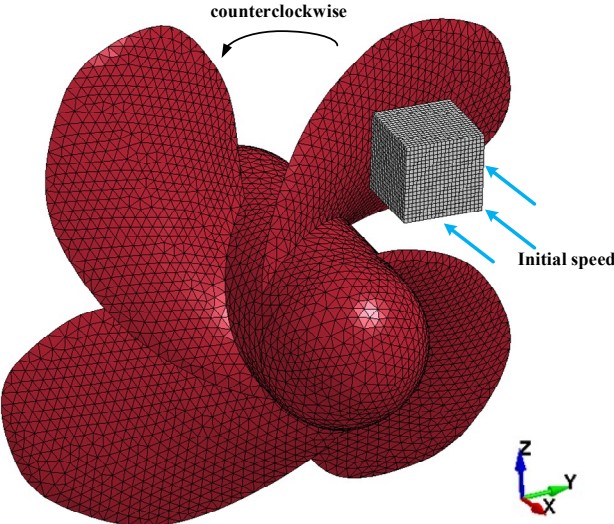

**Figure 4.** The numerical model of ice block impacting propeller.

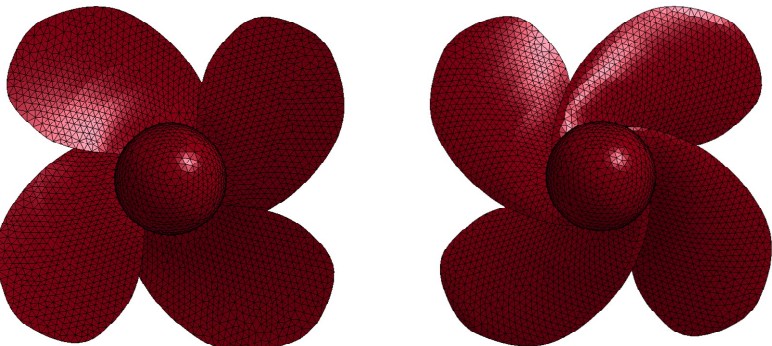

**Figure 5.** The model and mesh division of the propeller (Left: blade face; Right: blade back).

**Table 1.** The material parameters of the propeller.

| Items | Value |
|---|---|
| Density (kg/m³) | 7850 |
| Elastic modulus (GPa) | 200 |
| Poisson ratio | 0.3 |

The model of the ice block was made as cubic with a length of 66 mm, as shown in Figure 6. The ice block constituted hexahedral ice bulk elements and cohesive elements which are attached to bulk elements. An elastoplastic constitutive law of ice bulk elements is applied to simulate local ice-crushing failure against structures. The cohesive elements are used to simulate fracture failure of ice. The material of the ice block is shown in Table 2.

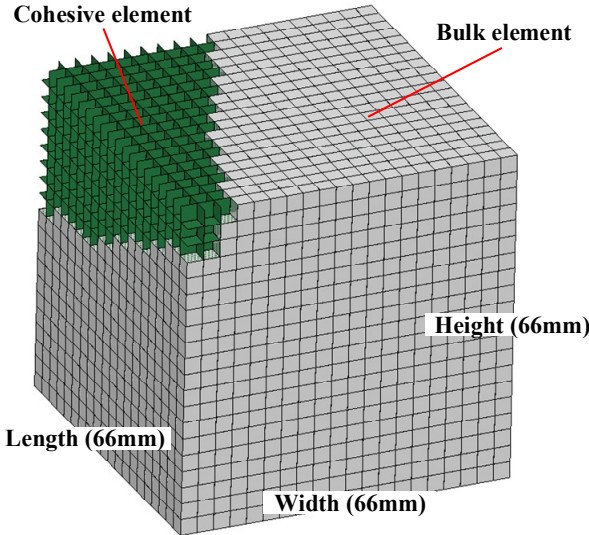

**Figure 6.** The model and mesh division of the ice block.

**Table 2.** The material parameters of ice bulk elements and cohesive elements.

| Bulk Elements | | Cohesive Elements | |
|---|---|---|---|
| Items | Value | Items | Value |
| Density (kg/m³) | 910 | Density (kg/m³) | 910 |
| Elastic modulus (GPa) | 5 | Elastic modulus (GPa) | 5 |
| Poisson ratio | 0.3 | $T_0$/Tensile strength (MPa) | 1.0 |
| Compressive strength (MPa) | 2.0 | $S_0$/Shear strength (MPa) | 1.0 |
| Crushed strain $\varepsilon_c$ | 0.35 | GIC/Fracture energy in mode I (J/m²) | 30 |
| Failure strain $\varepsilon_f$ | 0.5 | GIIC/Fracture energy in mode II (J/m²) | 30 |

All motions except rotation around x-axial are restricted. The motions of ice in all 6 dofs are set free without any constrain, which are analogue to ice collision with the blades. The built-in function of CONTACT-ERODING-NODES-TO-SURFACE in LS-DYNA is customized to detect the contact between the ice block and blades and thus calculate the contact force. The friction coefficient of the ice-blade is set to 0.2. The ice-ice contact is detected by the function of CONTACT-ERODING-SINGLE-SURFACE. The friction coefficient for ice–ice contact is set to 0.1. The explicit integral algorithm is used in LS-DYNA. The time step is determined by the minimum size of finite elements automatically and there is no need to define it by users.

## 4. Validation Study

### 4.1. Numerical Validation of Ice Material

The relationship of pressure and area curve is often used to validate the modelling of the ice material. The numerical calibration is based on the ice cone crushing tests carried out by Kim et al. [23]. The constant velocities (1 mm/s and 100 mm/s) are used to simulate the impact action between steel indenter and ice cone, as shown in Figure 7. The dimension of the steel indenter is about 400 mm × 400 mm and the diameter of the ice cone is 250 mm.

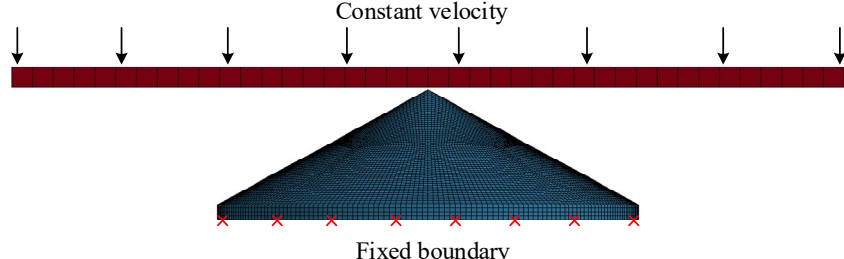

**Figure 7.** The model of ice-cone crushing tests.

Figure 8 gives the comparison between the numerical results of the presented method and the experimental results at different impact velocities. The simulation results show the same trend of the experimental results, i.e., the pressure curves decrease rapidly with the increase of nominal contact area, and the distributions of the scattered points are close to exponential form. The comparison also shows that the simulation results fit well with the experimental results. The difference is less than 20%, while the pressure-area curves between 800 to 1200 shows a consistent trend.

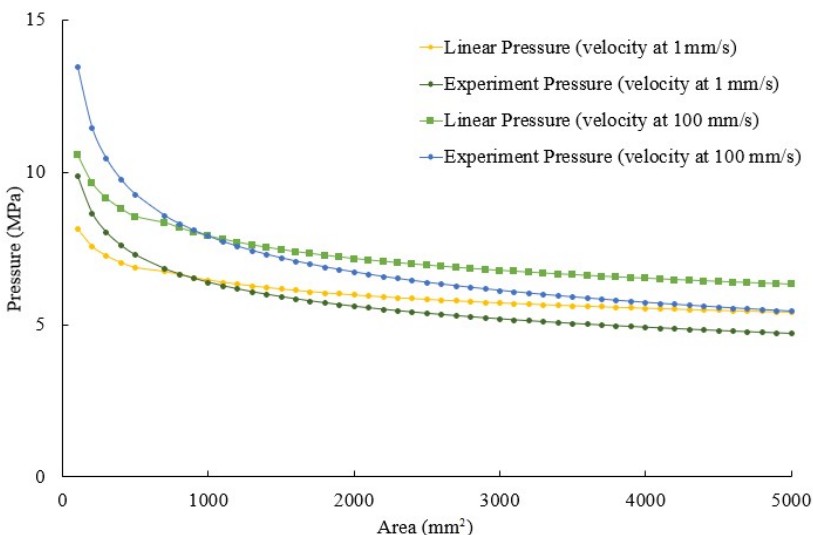

**Figure 8.** Comparison of pressure-area relationships between simulations and experiments.

### 4.2. Validation of Ice-Propeller Milling

Ice-propeller milling is also a kind of direct ice-propeller interaction scenario where the ice block is fixed and a rotating propeller will mill against ice with a certain speed. Since there is no public test data of ice-propeller collisions used for numerical simulation comparison, validation of the numerical ice-propeller milling process is presented as an alternative. Wang et al. [5] carried out a series of ice-propeller milling tests in the ice basin in Canada. The model test configuration is given in Figure 9a. In numerical simulation, the propeller model and ice block are set to be as the same as those used in the test as shown in Figure 9b. The other input parameters could refer to Wang et al. [25].

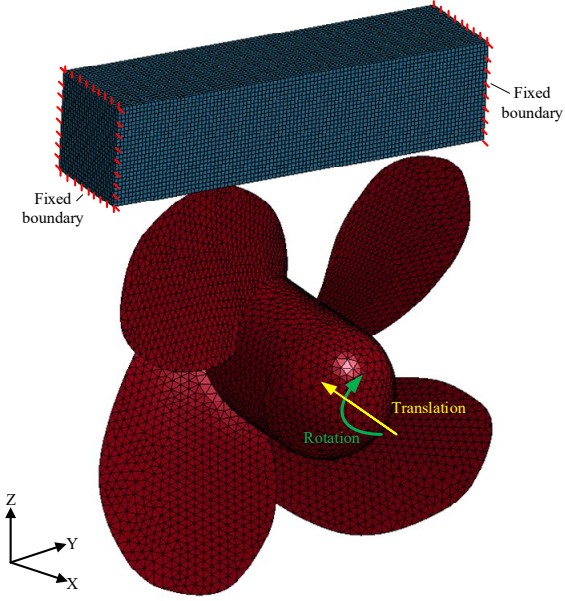

**Figure 9.** The configuration of ice-propeller milling.

The impact force and the moment from both numerical simulation and measurement are shown in the time domain with a duration of one second in Figure 10. Five peaks occur apparently for the two lines. Generally speaking, the simulation and the experimental results coincide well with each other. These peaks are extracted for comparison between the simulation and model test. Table 3 presents the comparison result for each peak value and mean value. The discrepancy between simulation and experiment is 0.4% for mean peak load and 9.1% for mean peak moments, which shows that the present simulation method is effective and reasonable.

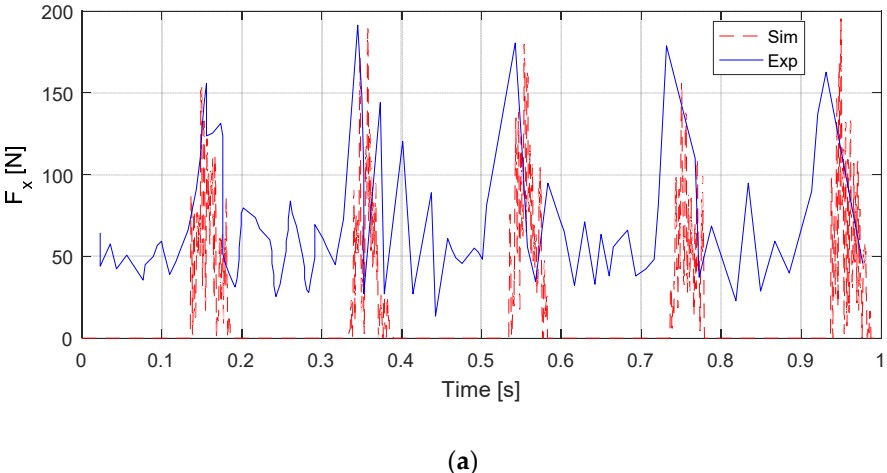

(**a**)

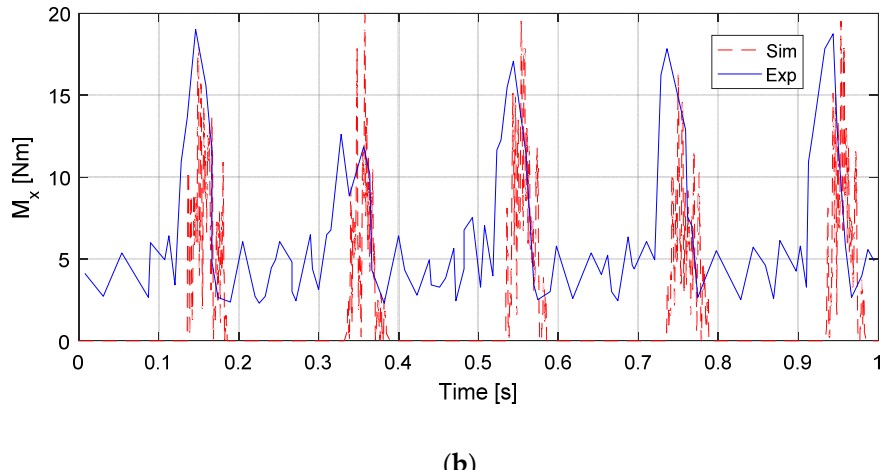

(**b**)

**Figure 10.** Simulated and measured ice loads during ice-milling process. (**a**) Force comparison between the simulation and the model test. (**b**) Torsional moment comparison between the simulation and the model test.

**Table 3.** Comparison of peak ice force and moment.

| Peak | Force | | | Moment | | |
|---|---|---|---|---|---|---|
| | Simulation(N) | Experiment(N) | Error | Simulation(Nm) | Experiment(Nm) | Error |
| Peak 1 | 151.6 | 155.9 | −2.8% | 17.95 | 19.02 | −5.6% |
| Peak 2 | 189.3 | 191.5 | −1.2% | 19.92 | 12.61 | 57.9% |
| Peak 3 | 179.8 | 180.5 | −0.4% | 19.52 | 17.07 | 14.4% |
| Peak 4 | 156.8 | 178.8 | −12.3% | 16.24 | 17.84 | −8.9% |
| Peak 5 | 195.4 | 162.7 | 20.0% | 19.51 | 18.75 | 4.1% |
| Mean | 174.6 | 173.9 | 0.4% | 18.62 | 17.06 | 9.1% |

## 5. Convergence Study

The simulation results may be sensitive to the meshing size of ice block. Therefore, we have to study the influence of meshing size in this section to show the reliability of present simulation method. Three kinds of meshing sizes, namely coarse, medium and dense, are selected. The sizes are d/80, d/120and d/150 respectively.

The simulations of ice block–blade collision with different ice-block meshing are performed in the time domain. The corresponding time series of axial force $F_x$ and moment $M_x$ are plotted in Figure 11. The collision process lasts around 0.05 second. Within the interaction, the ice block crushes against the surface of the blade and then leaves. It is clear from Figure 11 that the impact load increases in an oscillatory way and decreases afterwards. The amplitudes and mean forces are close for three cases with different meshing sizes. The mean, standard deviation and peak values for ice loads are calculated and summarized in Table 2. The relative errors of the simulated results with dense meshing are also included in brackets. It is found from Table 4 that there is no big difference between simulated results using dense mesh and medium. The differences regarding mean, standard deviation and peak forces are less than 0.4%, 1.5% and 3.0%. The simulation results converge well as the mesh size decreases. Considering the accuracy and efficiency of computation, it is suggested to use a medium mesh of ice block in the following simulations.

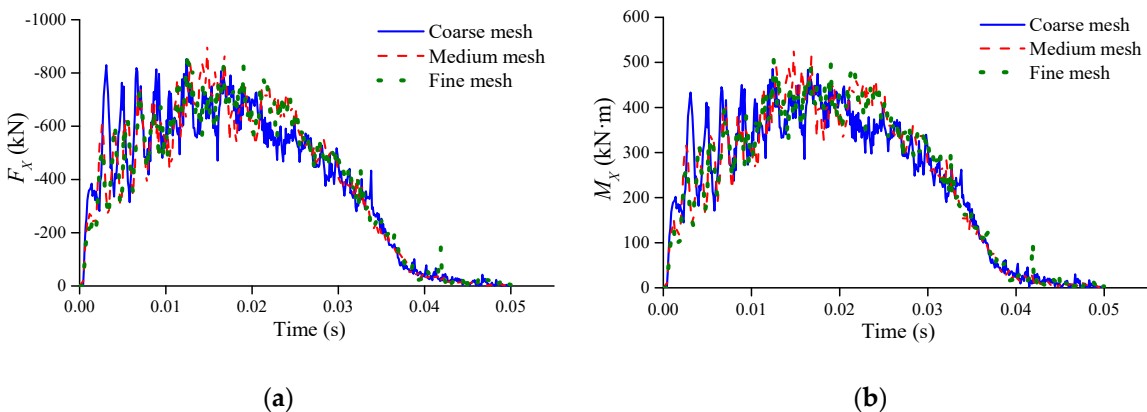

**Figure 11.** Time histories of ice loads for cases with different meshes. (**a**) The *Fx* with different meshes. (**b**) The *Mx* with different meshes.

**Table 4.** Comparison of ice loads for cases with different meshes.

| Mesh | $F_x$ (kN) | | | $M_x$ (kN·m) | | |
|---|---|---|---|---|---|---|
| | **Mean** | **StDev** | **Peak** | **Mean** | **StDev** | **Peak** |
| Coarse | -388 (0.8%) | 264 (2.7%) | -852 (3.3%) | 234 (1.7%) | 154 (4.3%) | 486 (4.3%) |
| Medium | -390 (0.3%) | 272 (1.5%) | -907 (3.0%) | 237 (0.4%) | 163 (1.2%) | 529 (4.1%) |
| Dense | -391 (-) | 268 (-) | -881 (-) | 238 (-) | 161 (-) | 508(-) |

Figure 12 presents three stress nephograms for an ice block at time instants of 0.01, 0.03 and 0.05 s. Clearly, we can observe the process of ice colliding and leaving off the blade. There is no obvious crack and fracture failure inside the ice block. The main failure mode is local crushing and shearing under high stain rate during the mutual interaction. Massive pulverized ice pieces tend to splash around the interaction surface due to crushing and shearing actions.

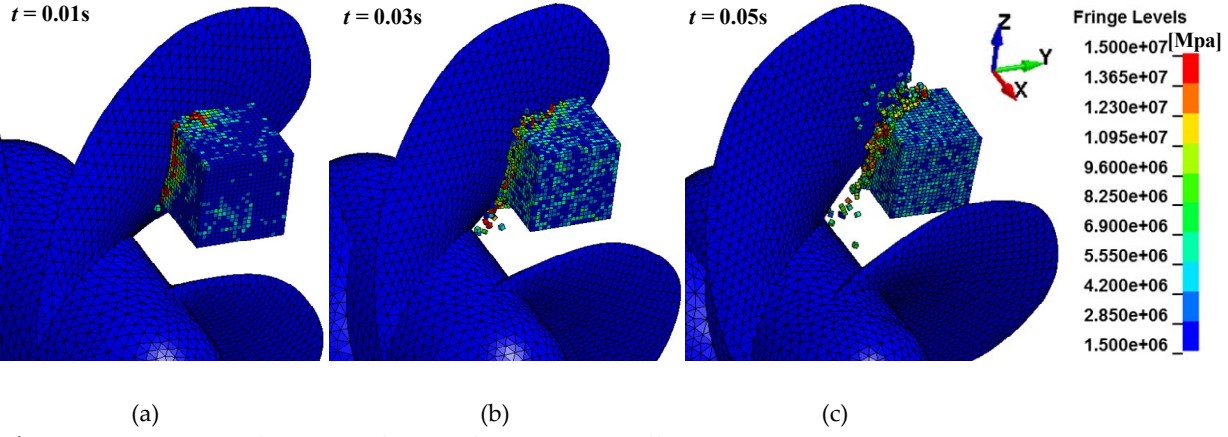

**Figure 12.** Stress nephograms during dynamic propeller–ice impacting process. (**a**) The ice impact simulation at 0.01s. (**b**) The ice impact simulation at 0.03s. (**c**) The ice impact simulation at 0.05s

A time series of ice speeds in three directions is shown in Figure 13. It can be seen that the ice speed in the x direction decreases quickly during interaction with the blade and becomes steady after leaving away. At the same time, the ice speeds in both y and z direction are no longer zero due to the ice-crushing effect. The resulting speed in the y direction is larger than z direction. The acceleration of ice block is relatively large during 0.01~0.02 s because of significant impact between ice and blade. As shown in Figure 11, the force increases to the maximum. As the ice speed in the x direction decreases and the ice speed in the y direction increase, the trend of the ice block leaving off

the blade becomes accelerative. When the ice starts to move away, the impact load drops with minor contact area.

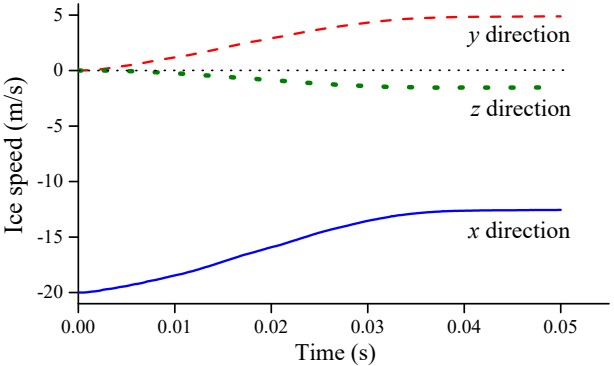

**Figure 13.** Variations of ice speed in three directions with time.

## 6. Parametric Study

This section focuses on the main factors which affect the simulation results. These factors include the rotational speed of the propeller, the rotational direction, the initial ice speed, the collision position and the contact area of the ice-propeller. The basic case has been given in Section 5. Only one factor is varied for analysis. The others are kept the same as those used in the basic case.

### 6.1. Effect of Rotational Speed

The influence of rotational speed of the propeller on impact load is then studied. The rotational speeds of 1rps, 1.5rps, 2rps, 2.5rps and 3rps are selected for analysis. It is taken as 2rps in the basic case. The propeller rotates anticlockwise in these cases. The time domains simulated ice load $Fx$ and moment $Mx$ with different rotational speeds are presented in Figure 14. It can be found that rotational speed affects dynamic ice loads significantly. As the rotation speed increases, the ice load decreases in general. The comparison among the maximum ice loads related to different rotational speeds is also shown in Figure 14. The results illustrate the ice load ($Fx$ and $Mx$) decreases with the increase of ice rotational speeds

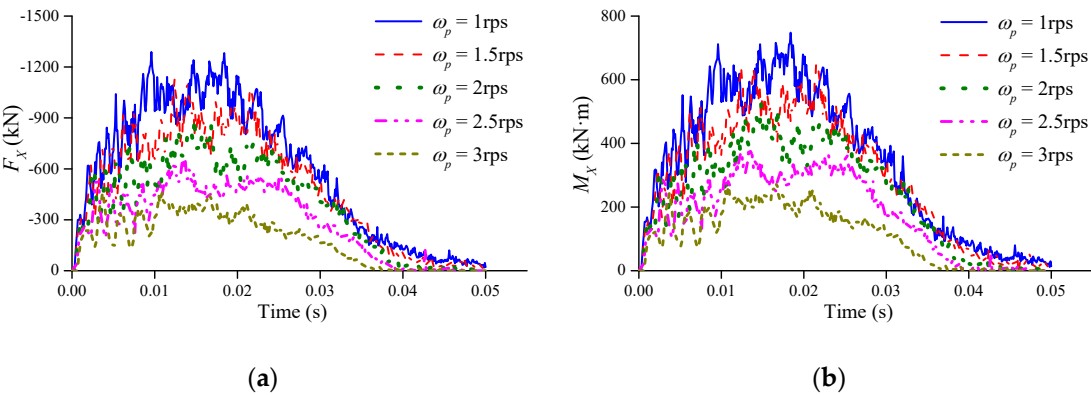

(**a**)　　　　　　　　　　　　　　　　　　　　　　　　　(**b**)

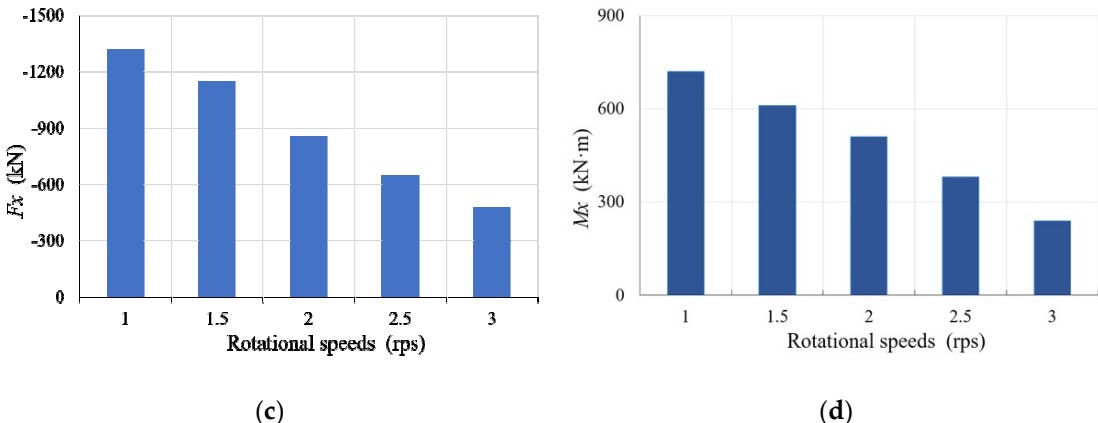

(**c**)                                                                 (**d**)

**Figure 14.** Time series of ice load *Fx* and *Mx* based on simulations with different rotation speeds. (**a**) The *Fx* with different rotation speeds. (**b**) The *Mx* with different rotation speeds. (**c**)The maximum *Fx* with different rotation speeds. (**d**) The maximum *Mx* with different rotation speeds.

Figure 15 gives the illustration of the ice block during interaction with the blades. When the propeller rotates anticlockwise, there exists an ice speed $V_p$ along the negative y-axis at any contact point. The normal component of the ice speed $V_p$ to the contact surface is defined as $V_{pc}$. The initial ice speed is $V_i$ in the minus x direction. Its normal component is expressed with $V_{ic}$. Then the relative crushing speed between the ice block and blade $\Delta V_c$ is equal to $V_{ic} - V_{pc}$. The higher $\Delta V_c$ is, the higher the impact load becomes. As the propeller rotates fast, the speed component $V_{pc}$ tends to ascend. This will lead to a decrease of the relative crushing speed $\Delta V_c$ and the resulting ice load.

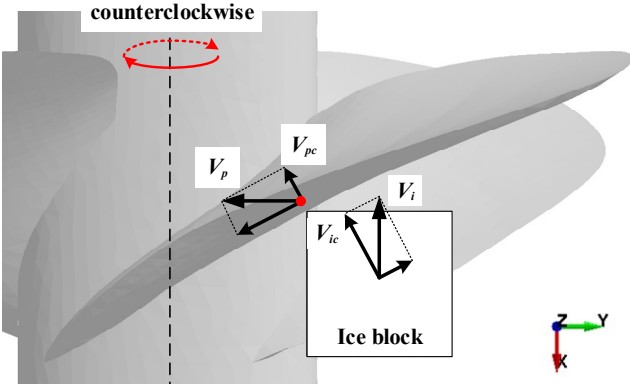

**Figure 15.** Schematic diagram of the speed during propeller–ice impacting process.

The time series of ice block velocity in the x direction under different rotational speeds of the propeller is shown in Figure 16. Clearly, the ice block speed drops quickly with slow rotation of the propeller. This is mainly attributed to severe compression and high impact load acting on the ice block. The ice speed drops slowly at a high rotational speed of the propeller.

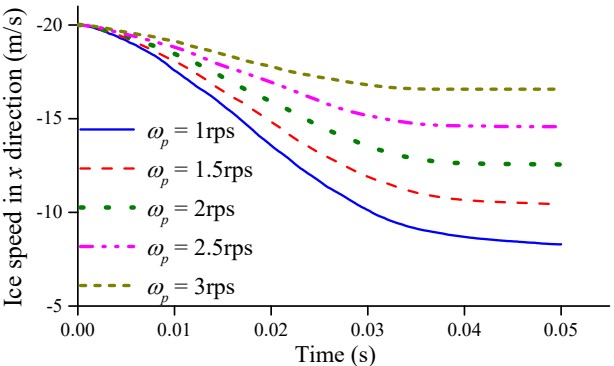

**Figure 16.** Time series of ice speed in the x direction based on simulations with different rotation speeds.

*6.2. Effect of Rotational Direction*

Rotational direction of the propeller is set to be anticlockwise in the basic study. This may affect the ice load as well and needs to be studied. The rotational direction is changed to clockwise with speeds of 1, 2 and 3rps. Figure 17 shows the time series of the ice load Fx and moment Mx from the simulated case with different rotational speeds. It is found that ice force increases to peak value quickly and decreases slowly to zero. The peak ice force tends to increase as the rotational speed increase when the propeller rotates clockwise. This is totally opposite to the phenomenon observed from the basic case. It is interesting to see that the rate of ice force declines with high rotational speed more than that with low rotational speed

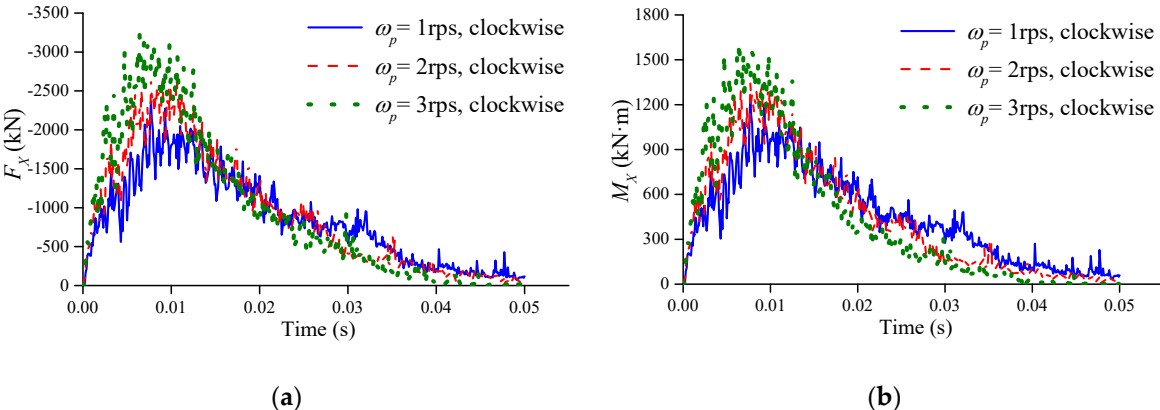

(**a**)          (**b**)

**Figure 17.** Time series of ice load *Fx* and *Mx* based on simulations with different rotation speeds for clockwise rotation. (**a**) The *Fx* with different rotation speed. (**b**) The *Mx* with different rotation speeds.

A schematic of ice block speed during collision with the propeller is presented in Figure 18. It could be seen that the relative crushing speed $\Delta V_c$ could be expressed by adding $V_{ic}$ and $V_{pc}$. The crushing interaction of ice and blades becomes more severe when the rotational speed increases, which results in an increasing of ice load at loading stage. A time series of ice block speed in the x direction under different rotational speeds is plotted in Figure 19, which shows that the higher the rotational speed is, the faster ice block speed drops. It should be noted that the ice speed changes from negative to positive at around 0.03 s for two cases with rotational speeds at 2 and 3rps, which means the ice block moves in an opposite direction under blade impact. However, movement direction of the ice block for the 1rps case is kept the same as the original. This contributes to a higher ice load compared to the other cases.

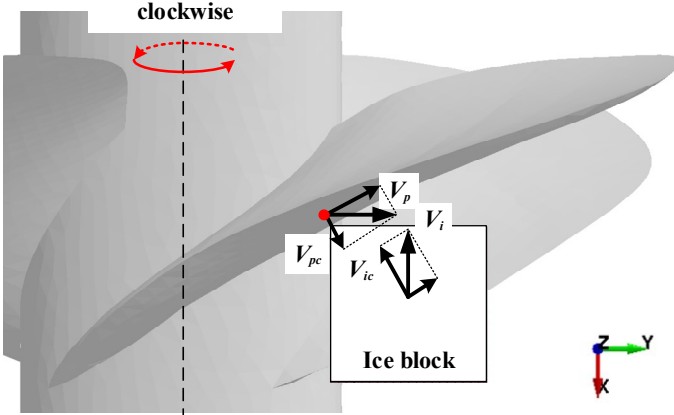

**Figure 18.** Schematic diagram of the speed during propeller–ice impacting process for clockwise rotation.

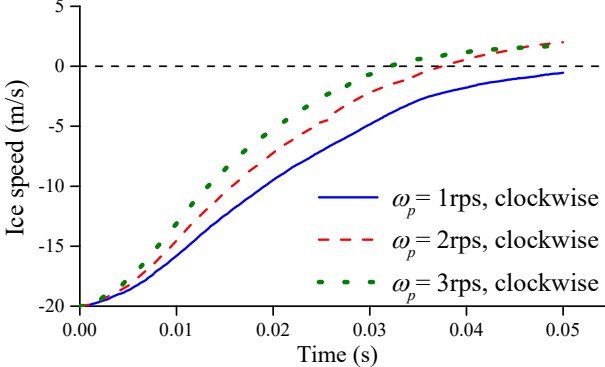

**Figure 19.** Time series of ice speed in the x direction based on simulations with different rotation speeds for clockwise rotation.

It shows a time series of ice loads under clockwise and anticlockwise rotation of the propeller with the same speed in Figure 20. It is observed that ice load is obviously higher in the clockwise case than anticlockwise. Moreover, the discrepancy increases as the rational speed increases.

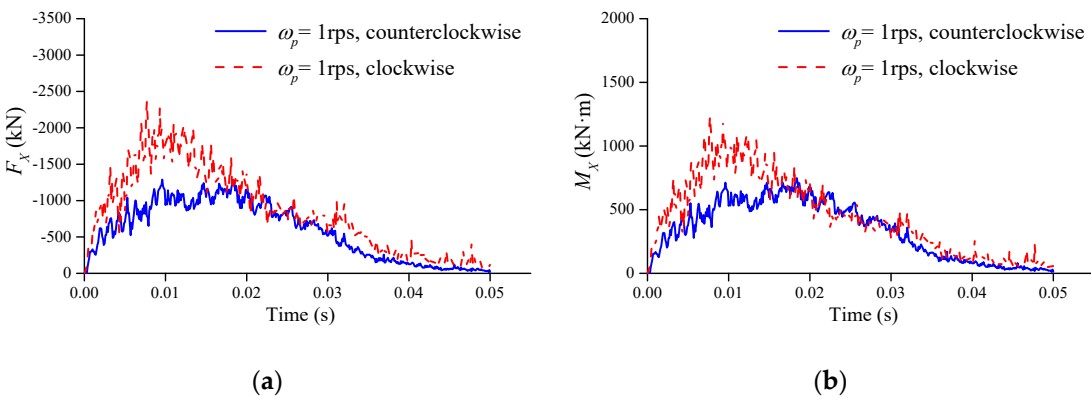

(**a**)    (**b**)

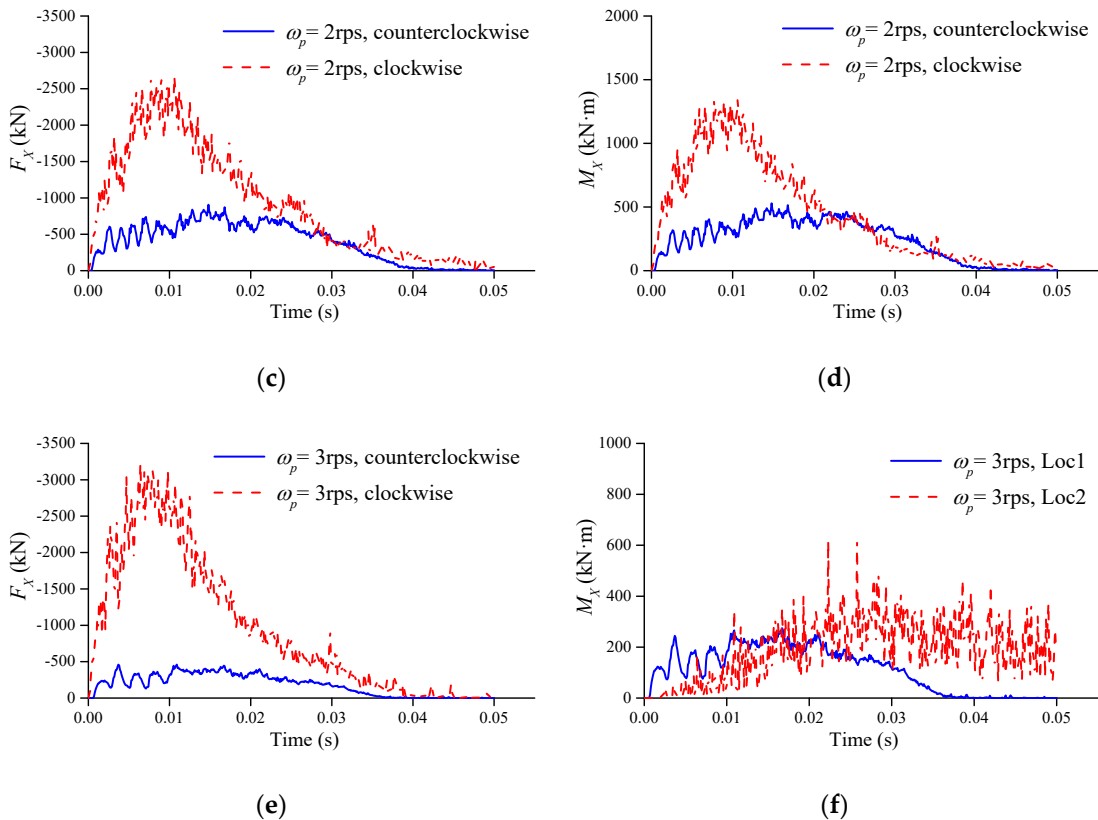

**Figure 20.** Time series of ice load *Fx* and *Mx* based on simulations with different rotation directions. (**a**) The *Fx* with 1rps. (**b**) The *Mx* with (**c**) The *Fx* with 2rps. (**d**) The *Mx* with 2rps (**e**) The *Fx* with 3rps. (**f**) The *Mx* with 3rps.

### 6.3. Effect of Initial Ice Block Speed

The effect of the initial ice speed on the ice load exposed to the propeller is studied using five initial speeds. They are 10, 15, 20, 25 and 30 m/s, where 20 m/s is applied in the basic case. Figure 21 presents the time series of ice loads *Fx* and *Mx* from simulations with different initial ice speeds. It is clear that the ice load is very sensitive to the initial ice speed. As the initial speed increases, the ice crushing impact on the blades becomes more and more intense, which leads to a significant increase of ice loads at both loading and unloading periods. It is noted that using very low initial ice speed such as 10 m/s will give very low ice load even down to zero. The comparison among the maximum ice loads related to different ice initial speeds is also shown in Figure 21. The results illustrate the ice load (Fx and Mx) increases with the ice initial speeds rapidly.

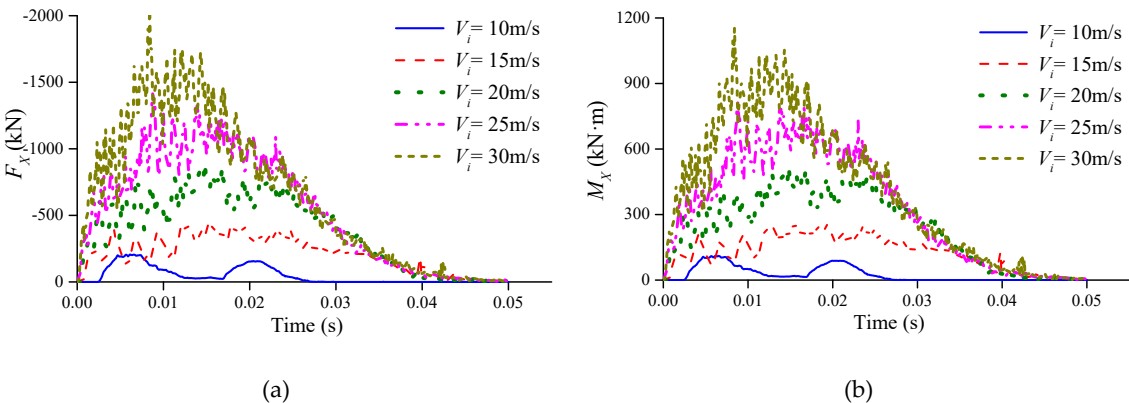

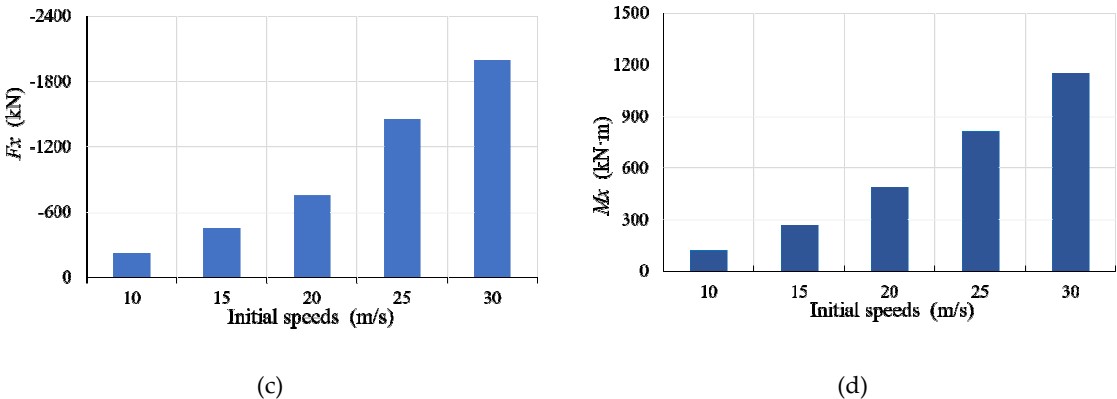

(c)                                        (d)

**Figure 21.** Time series of ice load *Fx* and *Mx* based on simulations with different ice initial speeds. (**a**) The *Fx* with different initial speeds. (**b**) The *Mx* with different speeds. (**c**)The maximum *Fx* with different initial speeds. (**d**) The maximum *Mx* with different initial speeds.

Figure 22 shows the trace of simulated ice forces for cases with different initial ice speeds. The rate of ice speed decline turns large as the initial speed increases. When the initial speed is set to 10 m/s, the ice speed is not affected too much. According to the simulation results, the relative ice and blade speed in the normal direction is calculated as $\Delta V_c = V_{ic} - V_{pc} \approx 0.187$ m/s, which is pretty low and thus could only produce ice load at a low level.

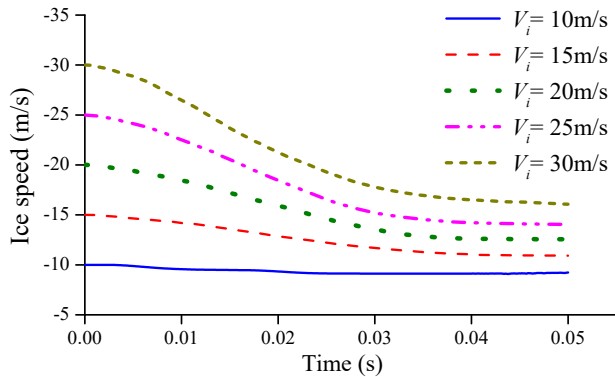

**Figure 22.** Time series of ice speed in the x direction based on simulations with different ice initial speeds.

### 6.4. *Effect of Contact Position*

The influence of ice contact position with the propeller on the ice load is studied. As shown in Figure 23, the contact position Loc1, which is marked by blade surface interaction with the ice block, is set as the initial position for the basic case. Then it is changed to Loc2, where blade edge collides with ice block. In this section, Loc2 is used with three rotational speeds of 1, 2 and 3rps in the simulation. Figure 24 presents the simulation results in the time domain, where the ice failure mode against the rotating propeller changes from crushing to milling. The ice bock is fractured clearly under the cutting effect of the blade edge. This effect could be observed more severely as the ice rotational speed increases. The ice bock is completely fragmented and cut in half in the condition of 3rps rotational speed at 0.05 s.

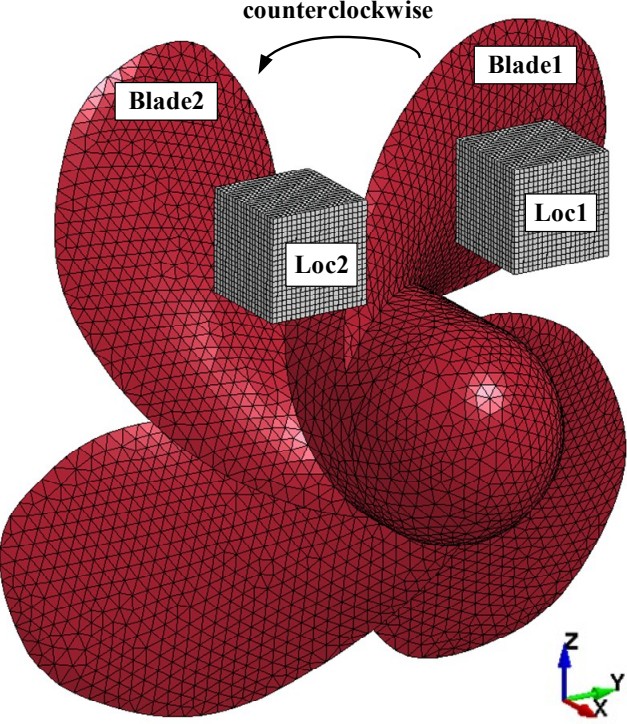

**Figure 23.** Numerical model of propeller-ice impacting at different collision locations.

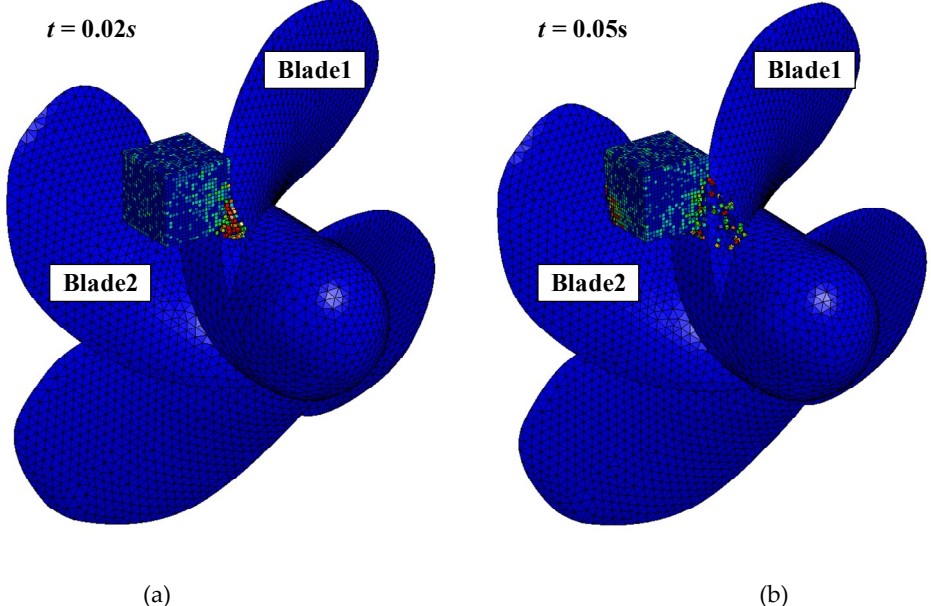

(a)                                                    (b)

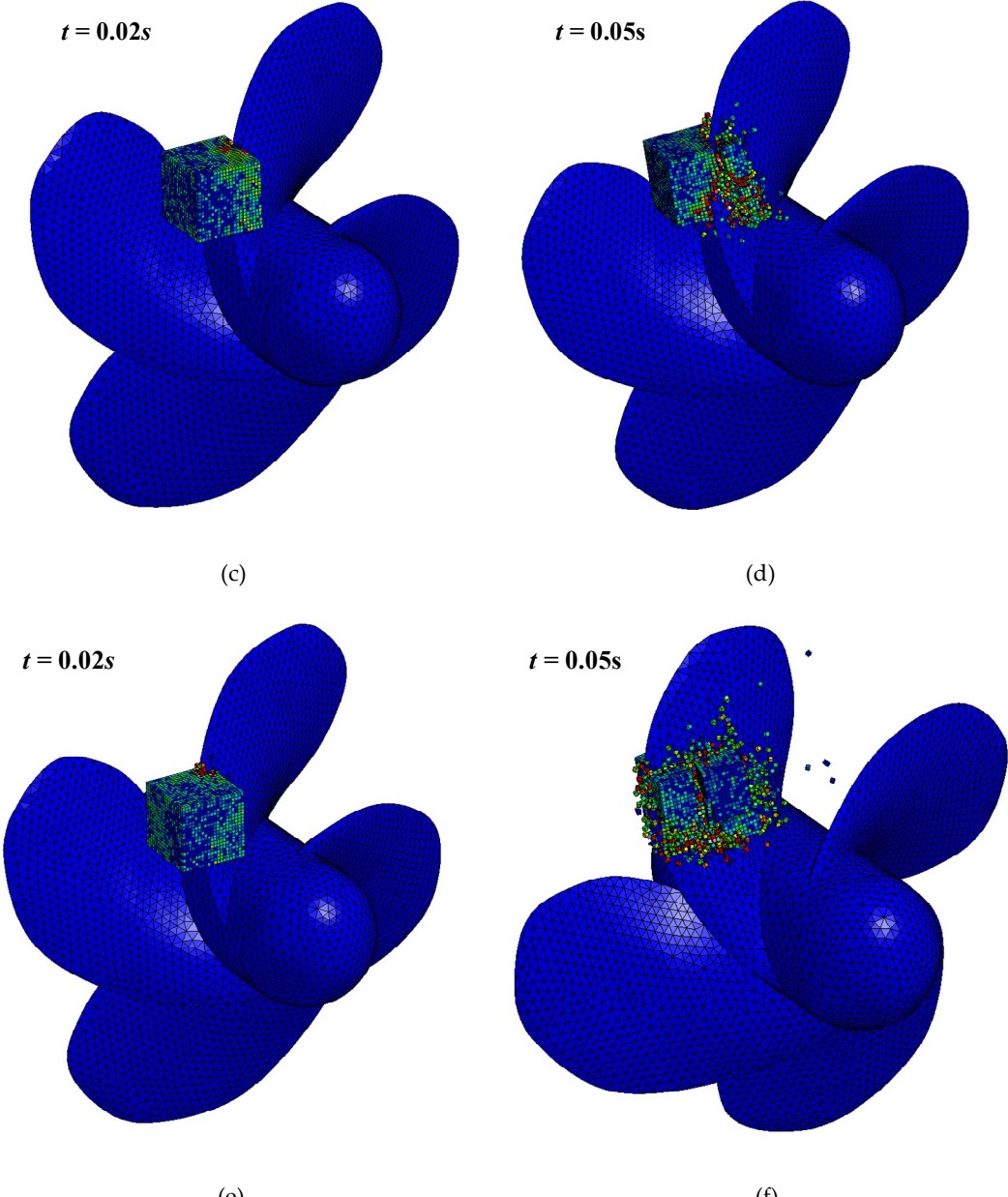

**Figure 24.** Stress nephograms of simulations for collision location at Loc2 with different rotation speeds. (**a**) The ice impact simulation at 0.02s with 1 rps. (**b**) The ice impact simulation at 0.05s with 1 rps. (**c**) The ice impact simulation at 0.02s with 2 rps. (**d**) The ice impact simulation at 0.05s with 2 rps. (**e**) The ice impact simulation at 0.02s with 3 rps. (**f**) The ice impact simulation at 0.05s with 3 rps.

Figure 25 shows the time series of ice loads Fx and moment Mx for cases with the same rotational speeds and different contact positions. We can find that ice loads acting on Loc1 are higher than that on Loc2 with the rotational speed of 1rps, which means that the crushing effect is dominant over the milling effect. As the rotational speed increases, the ice load on Loc2 augments rapidly. When the speed increases to 3rps, the ice load on Loc2 exceeds the ice load on Loc1. It should be noted that under the 1rps condition, the ice load on Loc2 rises again after 0.037 s, which seems abnormal. The ice block gets milled with the blade 1 at first. The ice block does not leave off the blade 1 before it crushes with a second blade due to slow rotation of the propeller. This could be observed clearly from Figure 24 a). The coming interaction with the blade 2 makes the ice load increase.

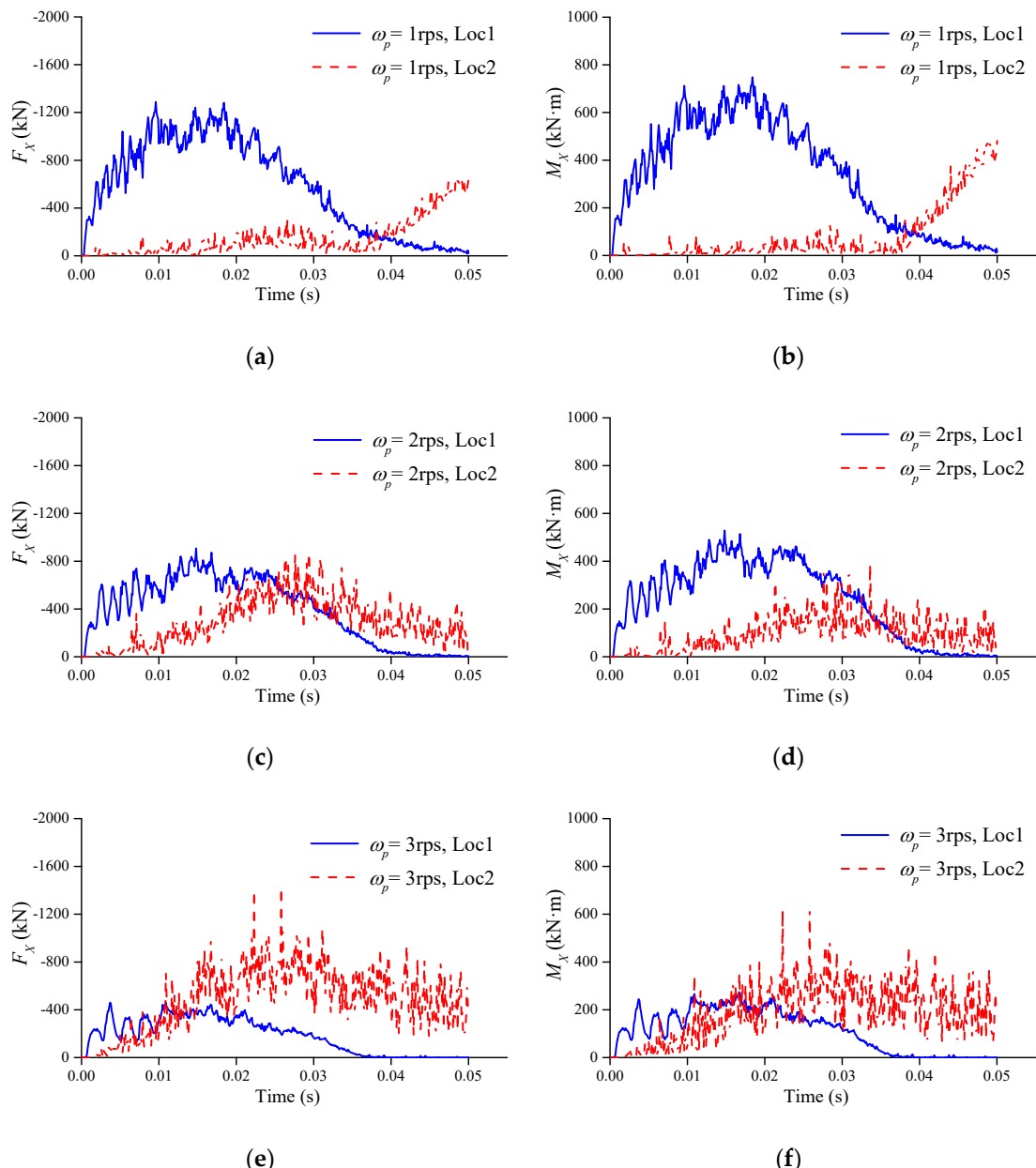

**Figure 25.** Time series of ice load Fx and Mx based on simulations with different collision locations. (**a**)The *Fx* with 1rps. (**b**)The *Mx* with 1rps. (**c**) The *Fx* with 2rps. (**d**)The *Mx* with 2rps. (**e**)The *Fx* with 3rps. (**f**)The *Mx* with 4rps

Figure 26 shows the evolution of ice block speed in the x direction at the location of Loc2 with different propeller rotation speeds. It is clear that ice block speed drops rapidly as the milling impact increases. The sudden drop of ice block speed for the 1rps case at 0.037 s is also found in Figure 26. This is also due to increased impact load when colliding with both blades 1 and 2.

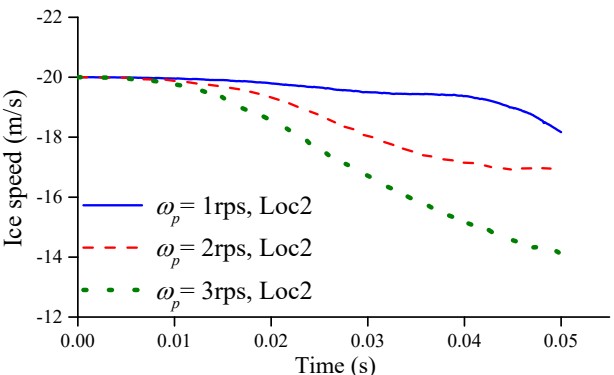

**Figure 26.** Time series of ice speed in the x direction based on simulations with different rotation speeds for collision location at Loc2.

## 6.5. Effect of Ice-Propeller Contact Area

In order to study the effect of ice-propeller contact area, we build up three types of ice block with different shape and the same weight as well as volume for simulation. Cube ice block is used in the basic study. Two other shapes are in pyramid as shown in Figure 27. Figure 28 shows the time series of ice loads $F_x$ and moment $M_x$ for cases using different ice blocks. The results show that the ice loads exposed to cube, paramid1 and pyramid 2 are in descending order. This means that the ice load increases as the contact area increases. Figure 29 shows the evolution of ice block speed in the x direction using different ice blocks. It is found that ice block speed drops rapidly as the contact area between ice and propeller increases. The stress nephograms for three ice block collisions at 0.04s are given in Figure 30, where the crushing phenomenon could be clearly observed.

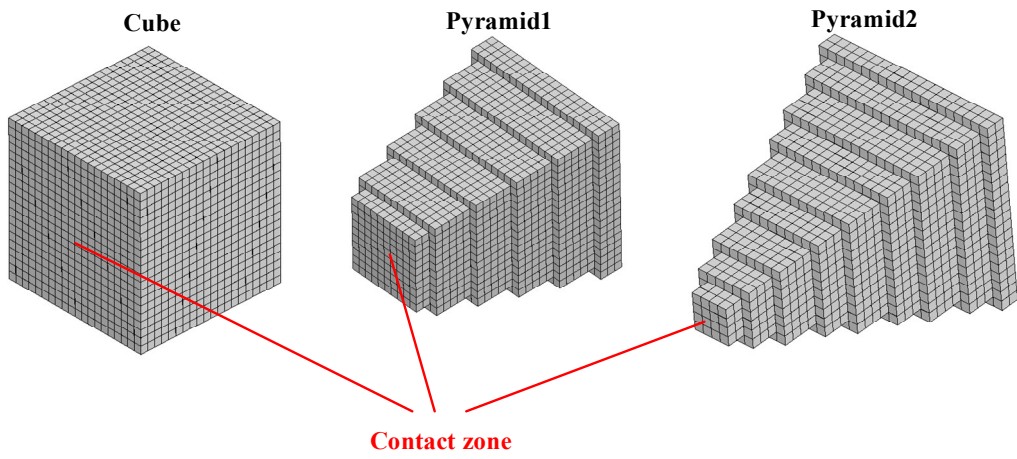

**Figure 27.** The models of three blocks with different shapes.

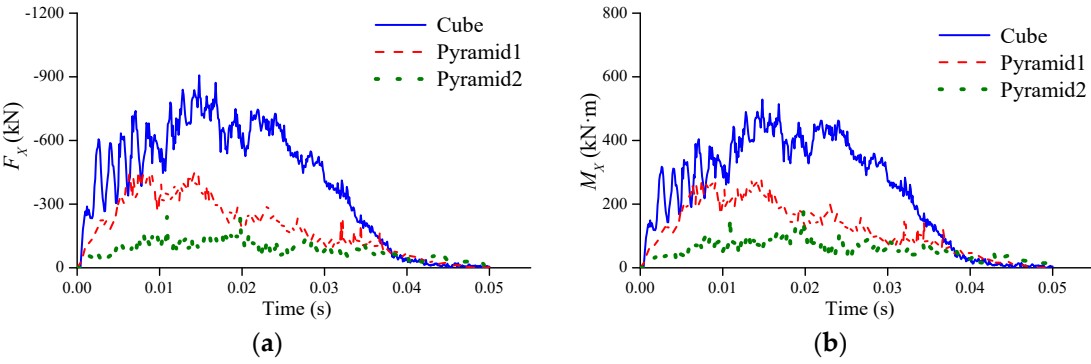

(a)　　　　　　　　　　　　　　　　　　　　　　　　　　　　　(b)

**Figure 28.** Time series of ice load *Fx* and *Mx* based on simulations with different ice blocks. (**a**)The *Fx* with different ice blocks. (**b**) The *Mx* with different ice blocks .

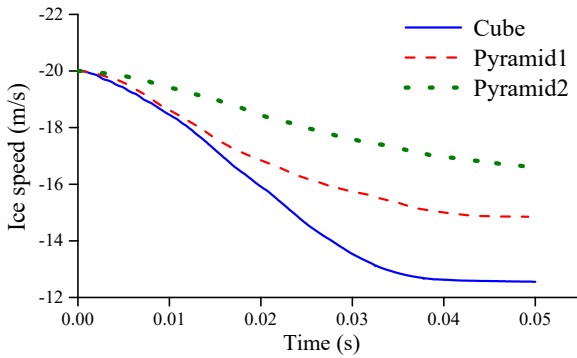

**Figure 29.** Time series of ice speed in the x direction based on simulations with different ice blocks.

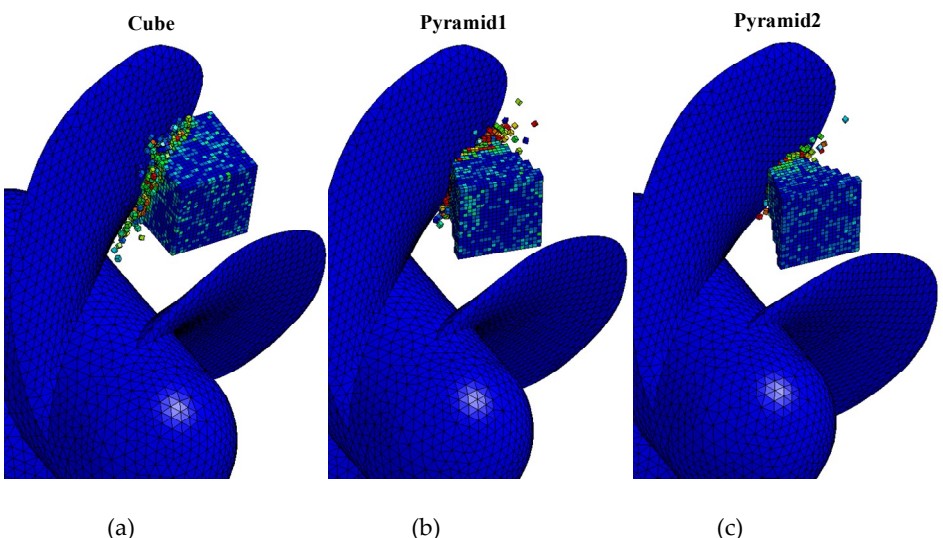

| (a) | (b) | (c) |

**Figure 30.** Stress nephograms of simulations for ice blocks with different shapes. (a) Impact simulation with cube ice block. (b) Impact simulation with pyramid1 ice block. (c) Impact simulation with pyramid2 ice block.

## 7. Conclusions

In this paper, a numerical tool based on cohesive element method is applied to simulate an ice-propeller collision process with the assistance of commercial finite element software LS-DYNA. The convergence study with respect to the meshing size of the ice block is conducted. Then, the effects of the rotation speed and the direction of the propeller, the contact position, the initial ice speed and the contact area of the ice on the resulting impact ice load are analyzed in detail. The main conclusions are summarized as follows:

(1) Different ice-failure modes could be observed in the simulation under the crushing and shearing action.

(2) Ice loads are influenced by using different ice meshing sizes to some extent. The dynamics of simulated ice loads become weak as the meshing size decreases.

(3) Based on the parametric study, the ice loads are significantly affected by all five factors considered. The dominant ice loads increase by decreasing rotational speed, increasing initial ice speed and contact area and changing rotational direction from clockwise to counterclockwise.

(4) When ice interacts with the edge of the blade under low rational speed of the propeller, an ice milling phenomenon occurs. It is possible that two blades will interact with the ice block simultaneously under slow rotation of the propeller. As the rotational speed increases, the ice load acting on the blade surface will exceed the milling load on the blade edge.

The simulation results show the present numerical method could capture the ice failure mode and calculate the ice-propeller collision loads. It is beneficial to understand the propeller–ice collision process. However, all conclusions have been drawn according to the present simulation tool. The simulation is expected to be refined in the future.

**Author Contributions:** Conceptualization, L.Z.; methodology, L.Z.; software, L.Z.; formal analysis, L.Z. and F.W.; investigation, L.Z.; resources, F.D.; data curation, S.D.; writing—original draft preparation, L.Z. and F.W.; writing—review and editing, H.Y.; visualization, Y.Z.; supervision, L.Z.; project administration, L.Z.; funding acquisition, L.Z. and S.D.

**Funding:** This research was funded by the National Natural Science Foundation of China, grant number 51809124, 51911530156; Natural Science Foundation of Jiangsu Province of China, grant number BK20170576; Natural Science Foundation of the Higher Education Institutions of Jiangsu Province of China, grant number 17KJB580006; State Key Laboratory of Ocean Engineering (Shanghai Jiao Tong University), grant number 1704 and 1807.

**Conflicts of Interest:** The authors declare no conflicts of interest.

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
