# Peer review of "Simulation of Ice-Propeller Collision with Cohesive Element Method"

_jmse, doi:10.3390/jmse7100349_

Round 1
Reviewer 1 Report
The paper presnts numerical model aplication to simulate important phenoman for ice-propeller colisions. Authors made both convergence and parametric study, but thay never compared obtaned results with laboratory data or tests made with the prototype. I wider if such calibration or validation will be possible . It will maderesults much more valuable and increase the quality of the manuscript.
Also, in my opinion the numerical model is not sufiecntly described. It will be good to add more inforations about the cohesive element method, which (i suppse) is used in LS-DYNA software. I wil suggest including governing equations used in the software and how boundary conditions were set.
Author Response
The paper presnts numerical model aplication to simulate important phenoman for ice-propeller colisions. Authors made both convergence and parametric study, but thay never compared obtaned results with laboratory data or tests made with the prototype. I wider if such calibration or validation will be possible . It will maderesults much more valuable and increase the quality of the manuscript.
Answer: This is a good suggestion. We accepted the comments and added a separate section 4 for validation study in the paper, which may improve the quality of the manuscript.
Also, in my opinion the numerical model is not sufiecntly described. It will be good to add more inforations about the cohesive element method, which (i suppse) is used in LS-DYNA software. I wil suggest including governing equations used in the software and how boundary conditions were set.
Answer: We accepted the comments and added the relevant content in “2 Numerical Model” in the paper.
Thank you for your valuable comments and suggestions!
Reviewer 2 Report
The authors should carry out a thorough revision of the article for grammar and consistency in the flow of information.
The authors only presented the propeller-ice interaction forces in FX. what about the forces and moments in the other coordinate directions?
The authors should provide evidence to the claim "The ice crushing, shearing and fracture failures are reproduced in the simulation which coincides well with observation from model tests.". A section should be added to the article to compare the ice failure modes predicted with the physical observation.
The authors should discuss the uncertainty in the simulated results and provide evidence on how uncertainty in various setup parameters affect the final results.
Author Response
The authors should carry out a thorough revision of the article for grammar and consistency in the flow of information.
The authors only presented the propeller-ice interaction forces in FX. what about the forces and moments in the other coordinate directions?
Answer: Fx and Mx are the dominant ice load to propeller induced by the impact of ice blocks. So in this paper, we focus on the key component for the study on simulation of ice-propeller collision. The other components are small compared these two values.
The authors should provide evidence to the claim "The ice crushing, shearing and fracture failures are reproduced in the simulation which coincides well with observation from model tests.". A section should be added to the article to compare the ice failure modes predicted with the physical observation.
Answer: Yes. Different kinds of ice failure phenomenon are very important. We are afraid that we have to reedit the sentence a little bit since the model test data and related failure pictures are difficult to obtain for us.
The authors should discuss the uncertainty in the simulated results and provide evidence on how uncertainty in various setup parameters affect the final results.
Answer: We agree with the suggestions completely. However, uncertainties came from too many factors. Therefore, we have to use different setups of one parameter while keeping others the same as the basic study. This has been done already in the paper. Sorry that we would like to keep the contents as it is.
Thank you for your valuable comments and suggestions!

Reviewer 3 Report
This reviewer cannot recommend publishing the paper in its current format. I believe there are some gaps in the results which means that the paper is of limited archival value.
For example, the authors present the CEM method, but do nothing to demonstrate the validity of the parameters chosen for the material model of ice. Could they present some results which compares the behaviour of their material with experiments on sea ice? For example, tensile or crushing loads for their material?
Similarly, there is no attempt to compare the results from the simulations with experiments by other authors. Although this may be difficult, some of the referenced matrial does contain enough information to get a comparison of the results.
A smaller point is that the results of the simulations could have been summarized in graphs, such as peak load against parameter.
Overall, the paper shows that the authors have successfuly managed to create some simulations using an interesting and potentially useful method within a commercial code, but there is nothing to demonstrate to readers of the paper that the results are accurate or meaningful.
Finally, the use of english throughout the paper need more careful attention to detail.
Author Response
This reviewer cannot recommend publishing the paper in its current format. I believe there are some gaps in the results which means that the paper is of limited archival value.
For example, the authors present the CEM method, but do nothing to demonstrate the validity of the parameters chosen for the material model of ice. Could they present some results which compares the behaviour of their material with experiments on sea ice? For example, tensile or crushing loads for their material?
Answer: We accepted the comments and add the Section 4 with “Validation Study” in the paper. Thank you for your valuable comments and suggestions!
Similarly, there is no attempt to compare the results from the simulations with experiments by other authors. Although this may be difficult, some of the referenced matrial does contain enough information to get a comparison of the results.
Answer: We accepted the comments and add the Section 4 with “Validation Study” in the paper.
A smaller point is that the results of the simulations could have been summarized in graphs, such as peak load against parameter.
Answer: This is a good suggestion. We accepted the comments and added the comparison of peak loads against rotational speeds and ice initial speeds in figure 14 and figure 21.
Overall, the paper shows that the authors have successfuly managed to create some simulations using an interesting and potentially useful method within a commercial code, but there is nothing to demonstrate to readers of the paper that the results are accurate or meaningful.
Answer: Yes. totally agree with the reviewer on this point. To acknowledge, we have no condition to conduct model test and compare simulated results with test data directly. Moreover, there are no public data available for comparison of simulation. The only way what we can do is to add Section 4 with Validation Study, where ice material is validated against model test data and ice-propeller milling process is also compared with model test. The comparison shows that this method is not bad according to our understanding.
Finally, the use of english throughout the paper need more careful attention to detail.
Answer: We accepted the comments and modified the grammar for the full text.
Thank you for your valuable comments and suggestions again!
Round 2
Reviewer 3 Report
I wish to thank the reviewers for the changes they made to the paper following the first review. I believe that they have addressed the reviewers' concerns as far as is possible, and I now think that the paper is acceptable for publication.
Author Response
Thank you very much for your good advise. We have reviewed the paper, and amended the language carefully.
Thank you again, and best wishes!